# Therapeutic Effects of Perilla Phenols in Oral Squamous Cell Carcinoma

**DOI:** 10.3390/ijms241914931

**Published:** 2023-10-05

**Authors:** Chia-Huei Lee, Yu-Hsin Tsao, Yui-Ping Weng, I-Ching Wang, Yao-Ping Chen, Pin-Feng Hung

**Affiliations:** 1National Institute of Cancer Research, National Health Research Institutes, Miaoli 35053, Taiwan; anita.tsao87@gmail.com (Y.-H.T.); 090406@nhri.edu.tw (Y.-P.C.); hdp91111@nhri.edu.tw (P.-F.H.); 2Department of Life Sciences, National Tsing Hua University, Hsinchu 30013, Taiwan; icwang@life.nthu.edu.tw; 3Department of Acupressure Technology, Chung Hwa University of Medical Technology, Tainan 71703, Taiwan; mbweng@gmail.com

**Keywords:** perilla, oral squamous cell carcinoma (OSCC), caffeic acid, rosmarinic acid, interleukin (IL)-1β, epidermal growth factor receptor (EGFR), gefitinib, cisplatin

## Abstract

The herbal medicine perilla leaf extract (PLE) exhibits various pharmacological properties. We showed that PLE inhibits the viability of oral squamous cell carcinoma (OSCC) cells. HPLC analysis revealed that caffeic acid (CA) and rosmarinic acid (RA) are the two main phenols in PLE, and reduced OSCC cell viability in a dose-dependent manner. The optimal CA/RA combination ratio was 1:2 at concentrations of 300–500 μM but had no synergistic inhibitory effect on the viability of OSCC cells. CA, RA, or their combination effectively suppressed interleukin (IL)-1β secretion by OSCC OC3 cells. Long-term treatment with CA and CA/RA mixtures, respectively, induced EGFR activation, which might cause OC3 cells to become EGFR-dependent and consequently increased the sensitivity of OC3 cells to a low dose (5 μM) of the EGFR tyrosine kinase inhibitor gefitinib. Chronic treatment with CA, RA, or their combination exhibited an inhibitory effect more potent than that of low-dose (1 μM) cisplatin on the colony formation ability of OSCC cells; this may be attributed to the induction of apoptosis by these treatments. These findings suggest that perilla phenols, particularly CA and RA, can be used as adjuvant therapies to improve the efficacy of chemotherapy and EGFR-targeted therapy in OSCC.

## 1. Introduction

*Perilla frutescens* (known as perilla), a plant commonly consumed in many regions of Asia, contains complex phytochemical constituents, such as flavonoids, volatile oils, unsaturated fats, triterpenoids, and phenols [1,2]. Because of their multiple aromatic rings and several hydroxyl groups, the phenols in *P. frutescens* cause the plant to exert considerable antioxidant effects [2,3]. Perilla also exhibits potent anti-inflammatory properties [4]. Chronic inflammation contributes to a microenvironment favoring the development and progression of various cancers, such as oral squamous cell carcinoma (OSCC) [5,6], renal cell carcinoma (RCC) [7], and colon cancer [8]. Because oxidative stress and inflammation are the causes of several pathological conditions, different parts of the perilla plant have long been used as traditional Chinese medicines to alleviate coughs, asthma, and constipation [2]. Hong demonstrated that perilla leaf extract (PLE) is a potent anti-SARS-CoV-2 agent that is capable of inhibiting SARS-CoV-2 replication [9]. In addition, treatment of PLE resulted in a reduction in virus-induced cytokine production in a human lung epithelial cell line, confirming the substantial anti-inflammatory properties of PLE. Several cell-based studies have reported the antitumoral efficacy of PLE in hepatocellular carcinoma [10], lung adenocarcinoma [11], and breast cancer [12]. Low dietary doses of perilla oil and PLE significantly reduced the risk of carcinogenesis in Apc^Min/+^ mice and rats with genetically and chemically induced colon cancer, respectively [13]. Perilla seed meal is rich in rosmarinic acid (RA), a plant-derived polyphenol. Using the lung adenocarcinoma cell line A549 in a test model, researchers showed that the RA-rich fraction of perilla seed meal exerts an antioxidant effect by scavenging tumor necrosis factor (TNF)-α-induced reactive oxygen species [14]. This fraction also suppresses the metastasis ability of A549 cells exposed to particulate matter [15]. Overall, the literature strongly indicates that compounds present in PLE are promising anticancer agents.

OSCC and RCC are associated with inflammation; the efficacy of PLE in treating these malignancies remains to be assessed. IL-1β is a key proinflammatory cytokine involved in carcinogenesis [16,17]. We previously demonstrated close associations between carcinogen exposure, IL-1β level, and oral cancer progression [5]. A high level of IL-1β was closely correlated with the malignant transformation of oral cells in a mouse model of OSCC. IL-1β promoted cell proliferation and dysregulated oncogenic signaling in keratinocyte cells derived from human oral mucosa dysplasia, suggesting a tumor-promoting role of IL-1β in the early stage of malignant transformation. Further investigation of the role of IL-1β in OSCC revealed a protumor molecular network involving autocrine IL-1β signaling and epidermal growth factor receptor (EGFR) signaling [18]. IL-1β transactivates EGFR by inducing C-X-C motif chemokine ligand 1 (CXCL1), thus activating C-X-C motif chemokine receptor 2 (CXCR2). Therefore, hyperactivated EGFR contributes to the tumor-promoting effect of IL-1β in OSCC. The role of IL-1β is more controversial in RCC than in OSCC. Although an in vitro study indicated that IL-1β is a pro-metastasis cytokine for RCC [19], we observed no association between IL-1β and tumor aggressiveness in RCC biopsies [20].

In the present study, we identified the anticancer phenols present in PLE and investigated the molecular mechanisms underlying their effects. We further explored the clinical potential of perilla phenols by assessing their use as an adjuvant therapy for reducing the doses of chemotherapy drugs and EGFR tyrosine kinase inhibitor (TKI) in cancer treatment.

## 2. Results

### 2.1. PLE Selectively Suppressed Viability of OSCC Cells 

Considering the anti-inflammatory activity of PLE [21], we investigated whether PLE exhibits selective cytotoxicity toward cancer types in which inflammation is an oncogenic factor. The effects of PLE on the viability of RCC 786-O and Caki-1 cells and OSCC OECM-1, TW2.6, and OC3 cells were assessed through the 3-(4,5-dimethylthiazol-2-yl)-2,5-diphenyltetrazolium bromide (MTT) assay. The treatment of the RCC cell lines 786-O (Figure 1a) with PLE concentrations up to 5 mg/mL for 48 h did not significantly affect their viability. By contrast, the viability of two OSCC cell lines, OC3 and TW2.6 (but not OECM1), tested in this study decreased significantly as the concentration of PLE increased from 1 L to 5 mg/mL. The susceptibility of the OSCC cells to PLE increased as their aggressiveness increased (Figure 1b–d). The viability of the less aggressive OECM1 cells was inhibited by only 18% after treatment with 5 mg/mL PLE for 2 days (Figure 1b). The most malignant cells, OC3 cells, were the most sensitive to PLE and were affected in a dose-dependent manner, exhibiting a considerable reduction in cell viability (to approximately 20%) after treatment with 4 mg/mL PLE for 2 days (Figure 1d). As shown in Appendix A, data on cell morphology indicated a correlation between sensitivity to PLE and the aggressiveness of OSCC cell lines. After the administration of 4 mg/mL PLE treatment over 48 h, OC3 cells shrank but OECM1 cells remained at approximately the same size and shape. In addition, the cell morphology of TW2.6 exhibited no evident change; however, the number of these cells decreased after incubation in media containing PLE for 48 h.

### 2.2. Phenols Present in PLE

The therapeutic potential of plant phenolic acids has attracted considerable attention [22]. To identify the phenols in PLE that effectively inhibit the survival of OSCC cells, we performed high-performance liquid chromatography (HPLC). As shown in Figure 2, caffeic acid (CA) and RA are the main phenolic components of PLE. PLE also contains luteolin and apigenin but in relatively small amounts. Thus, this study focused on the possible therapeutic effects of CA and RA in OSCC. 

### 2.3. Antiproliferation Effects of CA and RA on OSCC Cells

MTT assays were performed to assess the antiproliferation effects of CA and RA on TW2.6 and OC3 cells after 72 h of treatment. CA and RA exhibited IC_50_ values of 362.3 and 477 μM and 173.7 and 260.8 μM for TW2.6 and OC3 cells, respectively (Figure 3a,b). The viability of TW2.6 and OC3 cells decreased significantly as the CA concentration increased to 400 μM and 600 μM and RA concentration increased to 200 μM and 400 μM, respectively. Subsequently, we treated the cells with CA and RA combined at various ratios to identify the optimal combined antiproliferation effect of the two phenols. The viability of TW2.6 and OC3 cells decreased significantly when the cells were treated for 72 h with 300, 400, and 500 μM CA/RA combinations at ratios 1:1, 2:1, and 1:2 (CA to RA), respectively (Figure 3c,d). In TW2.6 cells, CA/RA combinations at ratios 1:1 and 1:2 inhibited cell viability more effectively than they did at a ratio of 2:1 (Figure 3c). In OC3 cells, the combination of CA and RA at a ratio of 1:2 most effectively inhibited cell viability at concentrations of 300 and 500 μM (Figure 3d). Combination ratios of 2:1 and 1:2 at a concentration of 400 μM reduced cell viability to a similar extent (Figure 3d). These data suggest that, in general, a CA/RA ratio of 1:2 is the most effective in reducing the viability of OSCC cells. Therefore, we determined the IC_50_ of the combination of CA and RA at a ratio of 1:2 for TW2.6 and OC3 cells. As shown in Figure 3e,f, this CA/RA mixture had an IC_50_ of 289.7 μM and 314 μM for TW2.6 and OC3 cells, respectively. With these combination parameters, CA and RA appeared to act antagonistically (combination index, 1.4) and additively (combination index, 1.0) in TW2.6 and OC3 cells, respectively. 

### 2.4. CA and RA Alone or in Combination Suppressed the Release of IL-1β from TNF-α-Induced or Non-Induced OC3 Cells

We previously demonstrated that the proinflammatory cytokine IL-1β is an oncogenic factor that promotes the malignant transformation of human oral cells and increases the aggressiveness of OSCC cells [5]. CA and RA exhibit anti-inflammatory effects [23,24]. We investigated whether CA or RA suppressed IL-1β secretion from OSCC cells. OC3 cells produce high levels of IL-1β [5]. Treatment with CA (200 μM) and RA (400 μM) alone or in combination significantly inhibited IL-1β secretion by OC3 cells in the presence or absence of TNF-α stimulation (Figure 4a). We analyzed intracellular IL-1β levels using Western blotting. As shown in Figure 4b, the differences in constitutive intracellular IL-1β levels between control cells and CA-treated or RA-treated cells were nonsignificant. Upon TNF-α stimulation, intracellular IL-1β levels were reduced by treatment with CA (120 μM) and RA (120 μM) alone or in combination. These results indicate that CA and RA alone or in combination inhibited constitutive or TNF-α-induced IL-1β secretion from OC3 cells, suggesting that these phenols can reduce the oncogenicity of the microenvironment in OSCC tumors. In the Western blot of glyceraldehyde-3-phosphate dehydrogenase (GAPDH), the minor band that migrates faster than the major band was one of the protein isoforms of GAPDH [25]. We used the intensity of the GAPDH major band as an internal control. 

### 2.5. CA and RA Alone or in Combination Induced EGFR Activation and Enhanced Antitumor Activity of Low-Dose Gefitinib

Secreted IL-1β transactivates EGFR in OSCC cells [18]. We hypothesized that CA and RA would affect EGFR activation because they suppress the secretion of IL-1β from OC3 cells (Figure 4a). CA (50 μM) treatment significantly enhanced the phosphorylation of EGFR in TNF-α-stimulated and unstimulated OC3 cells (Figure 5a). The combination of CA (50 μM) and RA (100 μM) also increased the abundance of phosphorylated EGFR in TNF-α-stimulated OC3 cells (Figure 5a). The level of total EGFR increased in CA-treated (50 μM) and RA-treated (100 μM) OC3 cells but not under the other test conditions (Figure 5b). The chronic activation of EGFR may lead to oncogenic dependence on the EGFR signaling pathway, thereby sensitizing cancer cells to EGFR TKIs such as gefitinib. We investigated whether long-term treatment with CA and RA increased the antitumor activity of gefitinib in OC3 cells. The IC_50_ of gefitinib was 8.3 μM for OC3 cells (Figure 5c). As shown in Figure 5d, OC3 cells were pretreated with CA (50 μM) and RA (100 μM) alone or in combination for 5 days and then cotreated with low-dose gefitinib (5 μM) for 3 days. Cell viability was assessed through MTT assays after treatment. Treatment of gefitinib (5 μM) for 3 days had no cytotoxicity toward the OC3 cells without pretreatment with CA and RA. Consistent with the effects on EGFR activation, incubation with CA (50 μM) alone or in combination with RA (100 μM) for 8 days significantly reduced the viability of low-dose gefitinib-treated OC3 cells (Figure 5e). RA (100 μM) alone did not significantly affect the viability of gefitinib (5 μM)-treated cells (Figure 5e). These results support the hypothesis that phenols present in PLE, particularly CA, would effectively improve the therapeutic efficacy of low-dose gefitinib in OSCC.

### 2.6. Chronic CA and RA Treatment Enhanced the Cytotoxicity of Low-Dose Cisplatin in TW2.6 Cells

Cisplatin is one of the most effective chemotherapeutic agents commonly used for treating OSCC; however, it may have severe adverse effects. Patients may benefit from a reduction in the dosage of cisplatin treatment. Cisplatin has an IC_50_ of 1.9 μM for both TW2.6 and OC3 cells (Appendix A). We examined whether chronic treatment with CA (50 μM) and RA (100 μM) increased the cytotoxicity of low-dose cisplatin in OSCC cells. Cells were grown in a medium containing CA (50 μM) and RA (100 μM) alone or in combination in the absence or presence of cisplatin (0.1 and 1 μM) for 7 days. We then used clonogenic assays to determine the survival of the OSCC cells. As shown in Figure 6, the viability of neither TW2.6 nor OC3 cells was significantly reduced after the 7-day treatment with low-dose cisplatin. CA (50 μM) alone did not affect the viability of the TW2.6 cells under any of the test conditions (Figure 6a). RA (100 μM) alone or in combination with CA (50 μM) significantly inhibited the viability of the TW2.6 cells with or without low-dose cisplatin treatment and increased the cytotoxicity of 1 μM cisplatin toward TW2.6 cells (Figure 6a). Treatment with CA (50 μM) and RA (100 μM) alone or in combination for 7 days markedly suppressed the survival of OC3 cells to less than 20% even in the absence of cisplatin but did not enhance the cytotoxicity of low-dose cisplatin toward OC3 cells (Figure 6b). These results indicate that long-term treatment with CA and RA alone or in combination can inhibit the survival of OSCC cells more effectively than does low-dose cisplatin and increase the cytotoxicity of low-dose cisplatin toward TW2.6 cells.

### 2.7. CA and RA Alone or in Combination Induced Apoptosis in OC3 Cells in a Time-Dependent Manner 

To investigate whether RA and CA inhibit the survival of OC3 cells at least in part by inducing apoptosis, we performed a time-course apoptosis assay through flow cytometry (Annexin V fluorescein isothiocyanate/propidium iodide staining). After treatment for 48 h, no apoptotic cells were detected. The apoptotic effect of CA (100 μM) and RA (200 μM) alone or in combination was noted in OC3 cells after treatment for 72 h (Figure 7a,b). And all three treatments exhibited comparable apoptotic effects. A minor proportion of necrotic cells (0.06–0.35%) was also detected. The apoptosis assay revealed that CA and RA alone or in combination induce apoptosis in OC3 cells.

## 3. Discussion

Plant-derived components have particular advantages over conventional synthetic molecules and are thus important in drug discovery. One of their advantages is conferred by their rigid structures that have been evolutionarily optimized and are therefore highly resistant to metabolism and degradation in vivo [26]. Perilla is an annual herb of the *Lamiaceae* family, which has been cultivated in China for more than 2000 years. Its leaves, stems, and seeds can be used as medicine and food. These parts exhibit various pharmacological activities, such as anti-inflammatory, anti-infection, antioxidant, detoxifying, and hepatoprotective activities [27]. Therefore, perilla has been extensively studied in the field of medicine. Many studies have investigated the anticancer activity of perilla and demonstrated its in vitro antitumor function in multiple cancers, including lung, colon, breast, and liver malignancies [27]. In this study, we explored the potential of perilla phenols for use in cancer therapy. The administration of natural compounds alone may be insufficient to suppress cancer progression; using them as an adjuvant to chemotherapy or target therapy may be a relatively viable treatment strategy. We demonstrated that PLE exerted anticancer effects on inflammation-associated OSCC cells (Figure 1). Furthermore, the sensitivity of OSCC cells to PLE was correlated with the levels of IL-1β secreted by these cells. In a previous study, we demonstrated that OSCC cells produced varying levels of IL-1β [5]. As indicated in Figure 1, the OC3 cells that secreted the highest levels of IL-1β (i.e., the most aggressive OSCC cells tested) were particularly sensitive to PLE treatment. In addition, the TW2.6 cells that produced lower levels of IL-1β were less sensitive to PLE treatment, whereas the OECM1 cells that produced the lowest levels of IL-1β among the OSCC cell lines exhibited little resistance to PLE treatment (Figure 1). Consistently, CA and RA alone or in combination effectively suppressed the production of IL-1β by TNF-α-stimulated and unstimulated OC3 cells (Figure 4). The effects of CA and RA on IL-1β levels have been observed in vivo; for example, CA reduces IL-1β levels in mice with acute and chronic cutaneous inflammation [28], and RA reduces IL-1β levels in rats with local and systemic inflammation [29]. Our results indirectly suggest that IL-1β inhibition by CA and RA contributes to the anti-OSCC effect of PLE and support the OSCC inhibitory capacity of CA and RA in vivo. We determined the CA/RA ratio of 1:2 as the optimal combination ratio at concentrations ranging from 300 to 500 μM. However, under these combination conditions, the CA/RA combination exhibited no synergistic effect on OC3 cells but exhibited antagonistic effects on TW2.6 cells, as indicated by the estimated confidence interval values (Figure 3c–f). The optimal combination ratio may vary depending on concentration. Therefore, identifying the optimal combination ratio at different concentrations may improve the antitumor activity of the CA/RA mixture in OSCC. The same combination exhibited antagonistic and additive effects on TW2.6 and OC3 cells, respectively, implying that the TW2.6 and OC3 cells have distinct tumor biology.

We previously demonstrated that blockage of the IL-1β/IL-1 receptor (IL-1R) axis using IL-1β-neutralizing antibodies significantly inhibited the constitutive activation of EGFR in OC3 cells [18]. However, in the present study, treatment with CA alone or combined with RA in OC3 cells with and without TNF-α stimulation suppressed IL-1β production and induced EGFR activation (Figure 5a). These results suggested that CA inhibits IL-1β production in a mechanism different from that of IL-1β- neutralizing antibodies. IL-1β-neutralizing antibodies block IL-1β/IL-1R signaling, thereby suppressing the downstream CXCL1/CXCR2 axis enabling the transactivation of EGFR in OC3 cells [18]. Two signals are critical to the synthesis and processing of IL-1β; the first signal activates nuclear factor-κB (NF-κB) for IL-1B transcription, and the other signal activates inflammasome, which in turn activates caspase-1 or caspase-8 to proteolyze pro-IL-1β to bioactive IL-1β [30]. To test whether CA and RA inhibited IL-1β expression by attenuating NF-κB activation, we performed a quantitative polymerase chain reaction (qPCR) to analyze the expression of *IL-1B* mRNA with TNF-α-stimulated TW2.6 and OC3 cells pretreated with CA, RA, or a mixture of CA and RA, as shown in Figure 4b. As shown in Appendix A, TNF-α treatment markedly increased the expression of *IL-1B* mRNA in TW2.6 and OC3 cells in the absence of CA and RA preincubation to approximately 2.3 and 3 times higher than their base levels, respectively. Treatment of CA and RA along or in combination significantly inhibited the TNF-α-induced expression of *IL-1B* mRNA in TW2.6, and the combined CA and RA treatment had no synergistic effect. Pretreatment of neither CA nor RA showed an inhibitory effect on the TNF-α-induced expression of *IL-1B* mRNA in OC3 cells. These results suggested that CA and RA have inhibitory effects on TNF-α-induced *IL-1B* transcription in TW2.6 cells, and this may be due to the suppression of the activation of NF-κB. In OC3 cells, both CA and RA have no effect on the TNF-α-induced activation of NF-κB. Accordingly, these results suggested that the inhibitory effect of CA or RA on IL-1β produced by OC3 cells may be exerted through the suppression of inflammasome activation. This finding indicates that the inhibition of IL-1β by CA, RA, or the combination of the two in OC3 cells may occur at the posttranslational level, namely through the activation of the inflammasome rather than at the transcriptional or translational level. Lee’s study showed that oral administration of caffeic acid phenethyl ester (CAPE) suppressed monosodium urate (MSU) crystal-induced caspase-1 activation and IL-1β production in mouse gouty arthritis models. CAPE is directly associated with the adaptor protein apoptosis-associated speck-like protein containing a caspase recruitment domain (ASC) in the nucleotide oligomerization domain-like receptor family pyrin domain containing 3 (NLRP3) inflammasome and blocks MSU crystal-induced NLRP3–ASC interaction, resulted in the inhibition of inflammasome activation and IL-1β production [31]. CA may suppress IL-1β secretion through the same mechanism as through which CAPE does. However, how CA induces EGFR activation in OC3 cells remains unclear and warrants further investigation. EGFR is frequently upregulated in various solid tumors including OSCC and non-small cell lung cancer (NSCLC). Chronic activation of EGFR may endow cancer cells with oncogenic dependence on EGFR signaling, thereby making them highly susceptible to EGFR TKIs such as gefitinib. Accordingly, NSCLC tumors harboring EGFR-activating mutations may be susceptible to treatment with gefitinib, whereas gefitinib exhibits limited efficacy against NSCLC with wild-type EGFR. We showed that 5 days of pretreatment with CA or CA/RA mixture increased the sensitivity of OC3 cells to low-dose gefitinib (Figure 5e). Because CA and CA/RA combinations respectively induced EGFR activation (Figure 5a), the pretreatment with CA or CA/RA combination for 5 days prompted OC3 cells to develop oncogenic dependence on EGFR pathways, thereby becoming sensitive to low-concentration gefitinib. Therefore, long-term treatment with CA or CA/RA combination may increase the therapeutic efficacy of EGFR TKIs in OSCC.

Chemotherapy is the primary treatment for OSCC. Although chemotherapy drugs are particularly toxic to cancer cells, they also damage normal cells that replicate regularly. Therefore, the administration of chemotherapeutic agents is often accompanied by severe adverse effects, such as vomiting, nausea, and neutropenia. These side effects sometimes prevent patients from receiving high doses of chemotherapy drugs to fight cancer as effectively as possible. Innovative approaches to improving the efficacy of chemotherapy have been developed; these approaches include targeted drug delivery systems, which use natural or synthetic polymers for the delivery of chemotherapeutic drugs to the tumor site, and chronochemotherapy, which refers to the administration of chemotherapy drugs in a time-specific manner [32]. We demonstrated that 7-day treatment with the CA/RA combination markedly enhanced the cytotoxicity of low-dose (1 μM) cisplatin toward TW2.6 cells (Figure 6b); chronic treatment with CA (50 μM) and RA (100 μM) alone or in combination substantially reduced the survival of OC3 cells in the absence of cisplatin (Figure 6b). These results suggest that CA and RA alone or in combination can be used as adjuvants to chemotherapy to considerably reduce the effective dose of cisplatin in OSCC. Moreover, patients with OSCC may benefit from the long-term administration of these perilla phenols.

In summary, we demonstrated that PLE contains phenols that can effectively induce apoptosis in OC3 cells and reduce the viability of OSCC cells. Furthermore, our findings suggest that the long-term oral administration of PLE or perilla phenols may improve the efficacy of EGFR TKIs and chemotherapy for OSCC and reduce the required drug dose. This highlights the clinical potential of perilla phenols in the treatment of OSCC.

## 4. Materials and Methods

### 4.1. Cell Culture

Cells were cultured in media as previously described [33,34]. Briefly, human oral squamous cell carcinoma (OSCC) OC3 and TW2.6 cell lines were grown in Dulbecco’s Modified Eagle Medium (DMEM, Cytiva, Marlborough, MA, USA) containing Keratinocyte serum-free medium (KSFM, Gibco, Waltham, MA, USA) and Ham’s F-12 Nutrient Mix (ThermoFisher Scientific, Waltham, MA, USA), respectively, in a 1:2 ratio, and supplemented with 10% fetal bovine serum (FBS, Gibco, Waltham, MA, USA). Human OSCC OECM1 and human renal cell carcinoma 786-O were grown in RPMI 1640 (Cytiva, Marlborough, MA, USA) medium containing 10% FBS (Gibco, London, UK). Human renal cancer cell line Caki-1 (also known as HTB 46) was maintained in Hyclone DMEM with high glucose (Cytiva, Waltham, MA, USA) containing 10% FBS (Gibco, Waltham, MA, USA). All the media contain 100 units of penicillin, and 100 μg of streptomycin/mL. All cell lines were cultured at 37 °C in a humidified atmosphere of 5% CO2.

### 4.2. Reagent Preparation

PLE (Cat. number 6216; approval number 022427 issued by the Ministry of Health and Welfare, Taiwan) was purchased from SunTen Pharmaceutical Co., Ltd. (Taipei, Taiwan) and prepared as previously described [9]. Briefly, PLE (25 mg/ mL stock solution) was sterilized using a 0.22 mm filter and stored at −20 °C less than 2 weeks. Caffeic acid (CA, Sigma-Aldrich, St. Louis, MO, USA) and rosmarinic acid (RA, Sigma-Aldrich, St. Louis, MO, USA) were dissolved with DMSO at concentration of 500 mM and 150 mM, respectively, and stored at −20 °C for less than 1 month. Cisplatin (Selleckchem, Houston, TX, USA) was dissolved with DMF at a concentration of 10 mM and gefitinib (Selleckchem, Houston, TX, USA) was dissolved with DMSO at concentration of 10 mM. 

### 4.3. Cell Viability (MTT) Assay 

Cells were seeded at a density of 5 × 10^3^ per well in a 96-well plate. The next day, cells were treated with the indicated concentrations of CA and RA alone or in combination. An appropriate amount of DMSO was also added to the control cells. After 72 h of treatment, the cells were incubated with a solution of MTT (3-(4,5-dimethylthiazol-2-yl)-2,5-diphenyltetrazolium bromide) (Sigma-Aldrich, St. Louis, MO, USA) for 2 h at 37 °C. Subsequently, the medium containing non-metabolized MTT was then aspirated, and 100 µL DMSO was added to solubilize the formazan. Colored formazan converted from MTT by viable cells was measured at 570 nm by a microplate reader (Bio-Rad, Hercules, CA, USA). Experiments were performed in triplicate.

### 4.4. High-Performance Liquid Chromatography (HPLC) Analysis of PLE

PLE was dissolved in methanol at a concentration of 3.3 g/mL for HPLC analysis. Four of the major active ingredients, including caffeic acid (Sigma-Aldrich, St. Louis, MO, USA), rosmarinic acid (Sigma-Aldrich, St. Louis, MO, USA), luteolin (MedChemExpress, Monmouth Junction, NJ, USA, and apigenin (MedChemExpress, Monmouth Junction, NJ, USA), of PLE were used as markers. The apparatus and chromatographic conditions of the HPLC system (Hitachi-High Technologies Corp., Tokyo, Japan) consisted of a binary pump, an ultraviolet detector (L-2400), a Primaide 1430 diode array detector and autosampler L-2200. System control and data analysis were processed with D-2000 Elite HPLC System software (version 2.0). The chromatographic separation was performed on an Athena C-18 WP (4.6 mm × 250 mm) using methanol–formic acid solution at 25 °C. The mobile phase consisted of methanol (solvent A) and 0.1% formic acid in water (solvent B) at a flow rate of 1 mL/min. Elution was performed with a linear gradient as follows: 0–1 min, A from 10% to 46%; 1–7 min, A from 46% to 50%; 7–12 min, A 50%; 12–22 min, A from 50% to 100% and 22–30 min, A from 100% to 10% [35]. The chromatogram was monitored at a wavelength of 330 nm during the experiment. The injection volume of each sample and standard solution was 5 to 50 μL. The HPLC mobile phase was prepared fresh daily, filtered through a 0.45 μm membrane filter and then degassed before injecting. 

### 4.5. Western Blot Analysis 

The cells were collected, washed twice with ice-cold phosphate-buffered saline (PBS), and then lysed with ice-cold cell lysis buffer (iNtRON Biotechnology, Gyeonggi-do, South Korea) containing freshly added protease and a phosphatase inhibitor cocktail (ThermoFisher, Waltham, MA, USA) on ice for 10 min. Samples were centrifuged at 13,000× *g* for 10 min at 4 °C. A sodium dodecyl sulfate (SDS) loading buffer was added to the supernatants, with subsequent incubation for 10 min at 100 °C. Aliquots of 20 μg of proteins were separated by SDS–polyacrylamide gel electrophoresis and electroblotted to polyvinylidene fluoride membranes. Membranes were blocked with 5% non-fat milk in PBS containing 0.05% Tween 20 (PBS-T) and then incubated overnight at 4 °C with indicated primary antibodies. After washed thrice with PBS-T, the membranes were incubated with horseradish peroxidase-conjugated secondary antibodies for 1 h at room temperature, followed by PBS-T wash three times and the bands were visualized using an ECL system (Merck, Rahway, NJ, USA). The intensities of the protein bands were quantified by ImageJ software (version 1.47). The protein expression of GAPDH was used as an internal control for normalization. 

### 4.6. Determination of IL-1β Levels in Culture Media

Cells were seeded at 3 × 10^5^ cells per well in a 48-well culture plate (Corning, New York, NY, USA) and grown in serum-free medium overnight. The next day, cells were pretreated with CA and RA alone or in combination with indicated concentrations for 8 h. The supernatants were collected and subjected to centrifugation at 10,000× *g* for 10 min at 4 °C. The secreted IL-1β in the supernatant was then quantified using an enzyme-linked immunosorbent assay (ELISA) kit for human IL-1β (Raybiotech, Norcross, GA, USA) following the manufacturer’s protocol [18]. Concentrations of measured IL-1β were normalized to the cell number determined in parallel.

### 4.7. Colony Forming

The cells were seeded at a density of 300 per well in a 6-well plate (Corning, New York, NY, USA). The next day, cells were incubated in medium containing CA (50 μM) or RA (100 μM) alone or in combination with indicated concentrations of cisplatin, changing the conditioned media every 48 h. An appropriate amount of DMSO was also added to the control cells. After incubation for 7 days, cells were washed with PBS and fixed with 4% paraformaldehyde for 40 minutes at room temperature. Colonies were counted after Giemsa staining as previously described [36].

### 4.8. Apoptosis Assay

The apoptotic effect of CA and RA was determined in OSCC OC3 cells, using a CoraLite 488-Annexin V and propidium iodide (PI) Apoptosis Kit (Proteintech, Rosemont, IL, USA) following the manufacturer’s instructions. Cells were treated with CA or RA alone or in combination with indicated concentrations, and control cells were treated with DMSO (0.02%). After 48 h and 72 h of incubation in conditioned medium, cells were washed with PBS and then resuspended in Annexin V binding buffer for Annexin V-FITC and PI labelling. Apoptotic cells were analyzed on a flow cytometer (FACScalibur, Becton Dickinson) by using an FITC signal detector and phycoerythrin emission signal detector. Attune^®^ NxT Software version 4.2 (Life Technologies, Carlsbad, NY, USA) was used for acquisition and data analysis.

### 4.9. Quantitative Real-Time PCR (qPCR) 

Total RNA was isolated using RNeasy Mini Kits (Qiagen, Hilden, Germany) according to the manufacturer’s instructions. The synthesis of cDNA from total RNA was performed with M-MLV reverse transcriptase (Promega, Madison, WI, USA). The resulting cDNAs were subjected to qPCR using the following primers: 5′-CCACAGACCTTCCAGGAGAATG-3′ (F) and 5′-GTGCAGTTCAGTGATCGTACAGG-3′ (R) for IL-1B. QPCR was performed with the FastStart Universal Probe Master Kit (Roche Applied Science, Madison, WI, USA). The gene expression level was normalized using GAPDH mRNA.

## Figures and Tables

**Figure 1 ijms-24-14931-f001:**
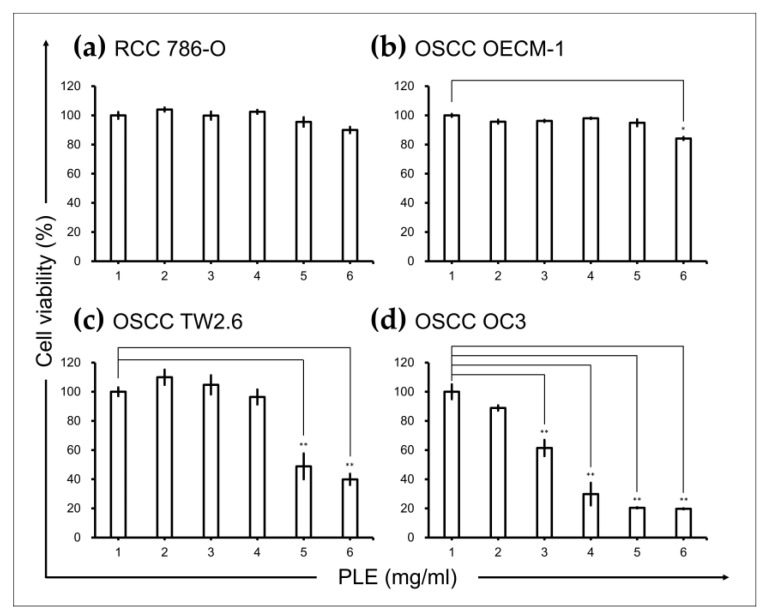
Effects of PLE on cancer cell lines. PLE exerted selective suppressive effects on oral squamous cell carcinoma (OSCC) cells. Renal cell carcinoma (**a**) and OSCC (**b**–**d**) cells were seeded in 96-well plates (density, 5 × 10^3^ cells per well), cultured for 16 h, and then treated with PLE at the indicated concentration for 48 h. Cell viability was assessed through the MTT assay. Data are presented as mean ± SEM of at least three independent experiments (* *p* < 0.05, ** *p* < 0.01). PLE, perilla leaf extract.

**Figure 2 ijms-24-14931-f002:**
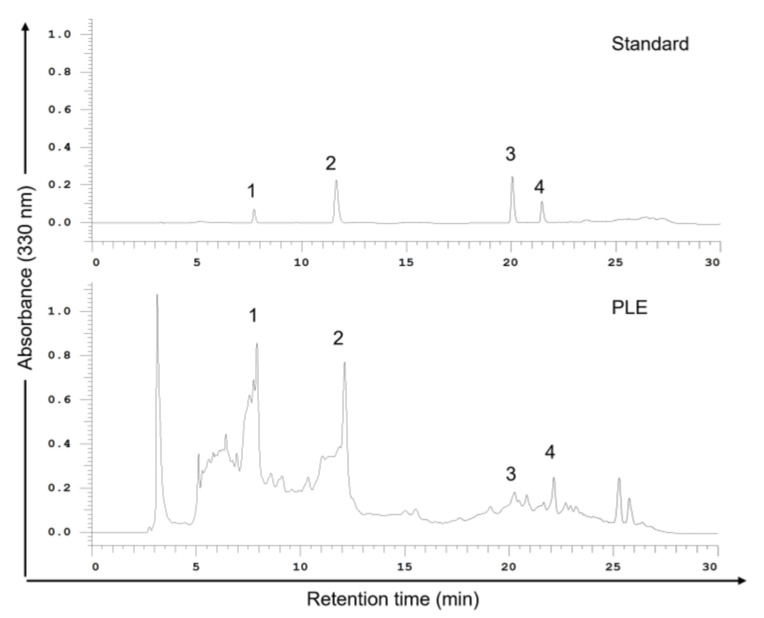
High-performance liquid chromatography (HPLC) chromatograms (330 nm) of an external standard PLE mixture. PLE (1 g) was dissolved in methanol (3 mL) at room temperature for 24 h. Samples were filtered through a 0.22 μM membrane before use, and 20 μL of the sample solution was injected into an HPLC instrument for filtration. The mobile phase comprised methanol and 0.1% formic acid in water; the flow rate was 1.0 mL/min. Peaks: 1, caffeic acid; 2, rosmarinic acid; 3, luteolin; 4, apigenin. PLE, perilla leaf extract.

**Figure 3 ijms-24-14931-f003:**
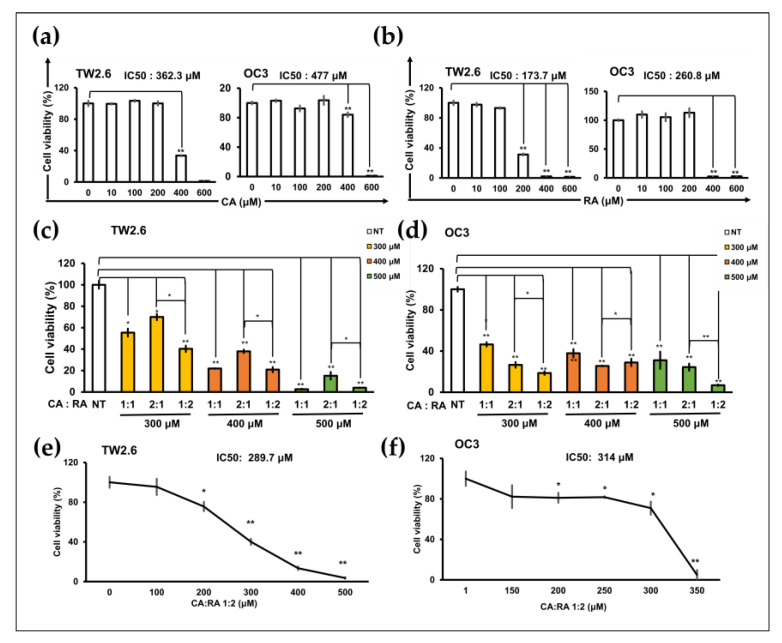
Caffeic acid (CA) and rosmarinic acid (RA) alone or in combination reduced the viability of TW2.6 and OC3 cells. (**a**,**b**) IC_50_ of CA (**a**) and RA (**b**) for TW2.6 and OC3 cells. (**c**,**d**) Optimal CA/RA combination ratio for reducing viability of TW2.6 (**c**) and OC3 (**d**) cells. (**e**,**f**) IC_50_ of CA/RA combination for TW2.6 (**e**) and OC3 (**f**) cells. Cells treated with 1:2 CA/RA combination at the indicated concentration. Cell viability was assessed through the MTT assay after 72 h treatment. The number of cells was normalized to that of dimethyl sulfoxide (DMSO) as the treated control. The color in the bar chart represents the final concentration of the combined mixture. NT, no treatment.Data are presented as mean ± SEM of at least three independent experiments (* *p* < 0.05, ** *p* < 0.01).

**Figure 4 ijms-24-14931-f004:**
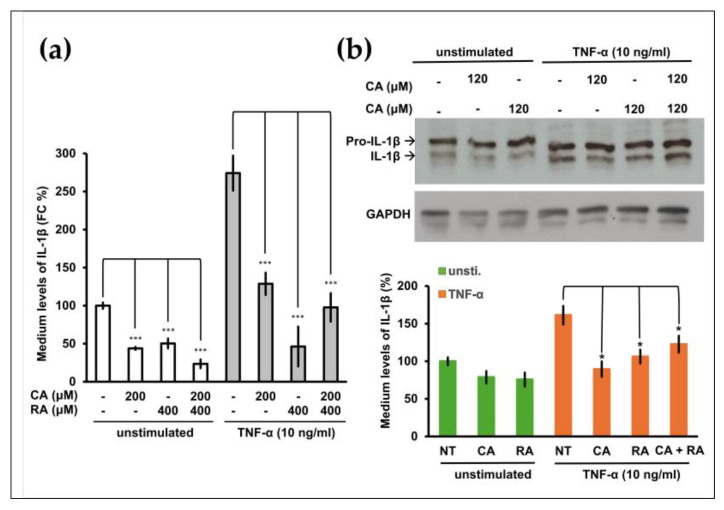
Effect of caffeic acid (CA) and rosmarinic acid (RA) on constitutive and tumor necrosis factor (TNF)-α-induced interleukin (IL)-1β production in OSCC OC3 cells. (**a**) OC3 cells were seeded in 48-well plates (density, 3 × 10^4^ cells per well), cultured for 16 h, and then pretreated with CA (200 μM) and RA (400 μM) alone or in combination for 8 h and then with or without TNF-α (10 ng/mL) for 16 h. The levels of IL-1β in the culture media were measured through ELISA. Data are presented in terms of the mean ± SD of at least three independent experiments (* *p* < 0.05, *** *p* < 0.005). (**b**) OC3 cells were pretreated with CA (120 μM) and RA (120 μM) alone or in combination in serum-free medium for 20 h and then with or without (unstimulated) TNF-α (10 ng/mL) for 4 h. Subsequently, intracellular pro-IL-β and IL-1β levels were analyzed through Western blotting. GAPDH was used as an internal control; the results of the Western blotting were quantified using ImageJ. Data are presented as mean ± SEM, *n* = 3. (* *p* < 0.05. *** *p* < 0.001).

**Figure 5 ijms-24-14931-f005:**
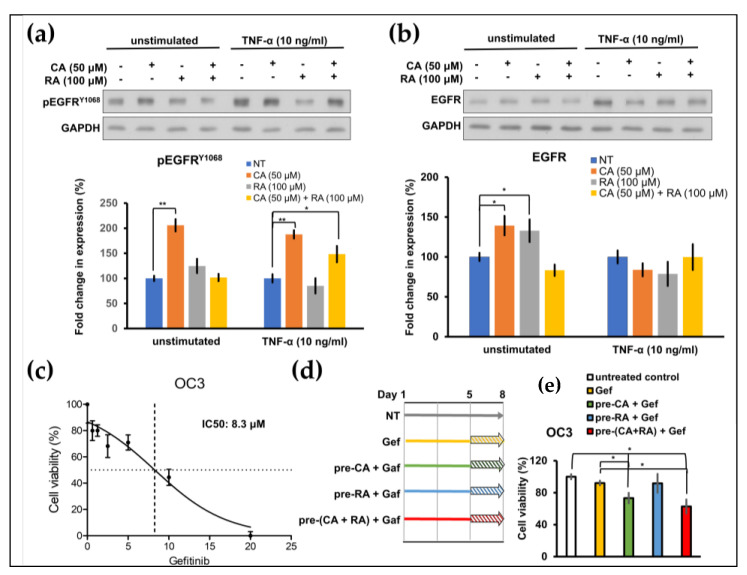
Effects of caffeic acid (CA) and rosmarinic acid (RA) on the constitutive and tumor necrosis factor (TNF)–α–induced phosphorylation of EGFR and the cytotoxicity of low-dose gefitinib toward OSCC OC3 cells. (**a**,**b**) OC3 cells were pretreated with CA and RA alone or in combination in serum-free medium for 20 h and then with or without (unstimulated) TNF–α (10 ng/mL) for 4 h. Subsequently, cellular protein lysates were harvested and subjected to Western blotting using antibodies against phosphorylated EGFR^Y1068^ (pEGFR^Y1068^, **a**) or EGFR (**b**). Quantification was performed using ImageJ. Data are presented as mean ± SEM, *n* = 3. (**c**) Dose–response curve for OC3 cells treated with gefitinib. Cell viability was determined through the MTT assay after treatment with the indicated concentration of gefitinib for 72 h. (**d**) Schematic of experimental protocol of (**e**). Cells were pretreated with CA (50 μM, pre–CA) and RA (100 μM, pre-RA) alone or in combination (pre–CA + RA) for 5 days and then with gefitinib (5 μM) for 3 days or left untreated. Cell viability was determined through the MTT assay after gefitinib treatment (* *p* < 0.05, ** *p* < 0.01).

**Figure 6 ijms-24-14931-f006:**
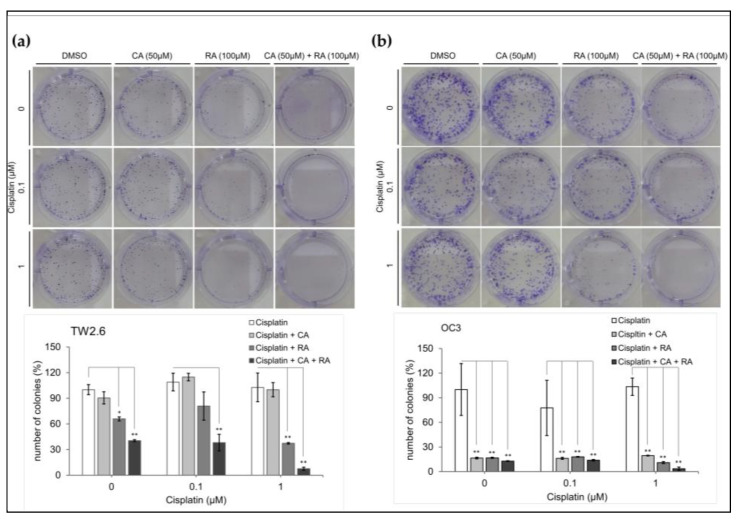
Effect of caffeic acid (CA) and rosmarinic acid (RA) alone or in combination on the cytotoxicity of low-dose cisplatin toward OSCC cell lines. Cell survival was assessed through clonogenic assays. (**a**) TW2.6 and (**b**) OC3 cells were seeded in 6-well plates (density, 300 cells per well) and cultured in medium containing CA and RA alone or in combination with or without cisplatin at the indicated concentration. Number of colonies counted after incubation for 7 days using MetaMorph software (version 7) and normalized with control (DMSO) colony numbers. Upper section: representative images of colonies grown for 7 days under indicated conditions. Bottom section: data present mean ± SEM, *n* = 3. Statistical differences between cells treated with CA and RA and control cells were analyzed using Student’s *t* test; * *p* < 0.05 and ** *p* < 0.01.

**Figure 7 ijms-24-14931-f007:**
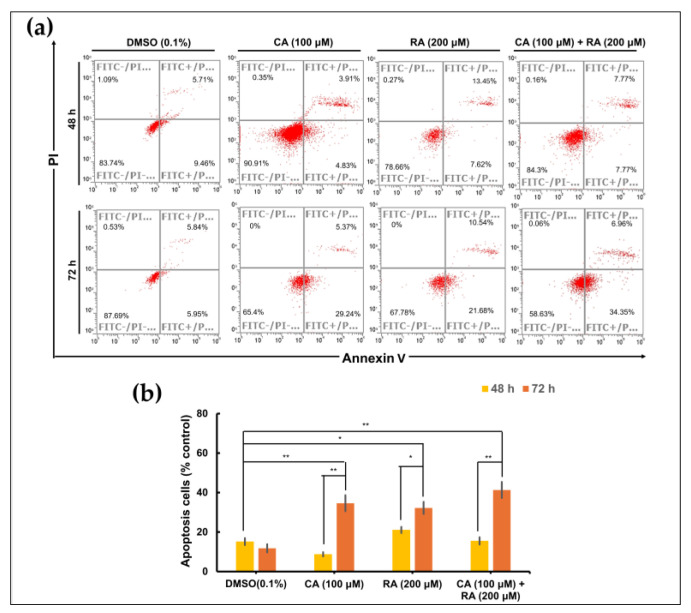
Apoptotic effects of caffeic acid (CA) and rosmarinic acid (RA) alone or in combination on OC3 OSCC cell line. (**a**) Flow cytometry analysis with Annexin V—PI staining was performed to evaluate the percentage of apoptotic cells in the OC3 cells treated with CA (100 μM) and RA (200 μM) alone or in combination for 48 or 72 h. Control cells cultured in medium containing DMSO (0.1%) only. (**b**) Analysis on cell apoptosis results of OC3 cells. Data are presented as mean ± SEM, *n* = 3. The significance of the difference between the 48 h and 72 h treatments was calculated using Student’s *t* test; * *p* < 0.05 and ** *p* < 0.01.

## Data Availability

No new data created.

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
