# Peer review of "Therapeutic Effects of Perilla Phenols in Oral Squamous Cell Carcinoma"

_ijms, 2023, doi:10.3390/ijms241914931_

Round 1

Reviewer 1 Report

In this manuscript, the authors studied the antitumorigenic properties of the herbal medicine Perilla leaf extract on oral cancer cells. The authors identified that PLE inhibited the growth of aggressive oral cancer cell line OC3. Further analysis of the PLE extract, the authors determined that the active principles in PLE are caffeic and rosmarinic acid. Using these active compounds and their combination, the authors showed that these compounds prevent the transactivation of EGFR by inhibiting IL-1b and thus inducing apoptosis. Even though the study had provided a new understanding of the action of this herbal medicine on oral cancer. The authors are requested to address all the following major comments.

1.      The authors claim that “ the viability of all three OSCC  cell lines tested in this study decreased significantly as the concentration of PLE increased  from 1 mg/mL to 5 mg/mL.” The viability graph shows only TW2.6 and OC3 showed a significant reduction in viability compared to OECM -1. The authors need to rewrite this section. In addition, the authors need to explain why PLE is less sensitive to the OECM-1 cell line. As supplementary, the authors are requested to provide representative images of cell morphology before and after treatment with PLE for all the OSCC cell lines. This information will reiterate the authors’ claim “that susceptibility of OSCC cells to PLE increased as their aggressiveness increased”.

2.      In line 126, the authors mention cell line CO3. “ In CO3 cells, the combination of CA and RA at a ratio of 1:2 most effectively inhibited cell viability at concentrations of 300 and 500 μM” The reviewer could not find much information on this specific cell line in the materials and method section. The authors are requested to provide more information regarding the CO3 cell line.

3.      In Figure 4B, why are there two bands for GAPDH blotting instead of a single band? The authors need to explain why there are two bands.

4.      The reviewer is keen to know whether the inhibition of IL-1b by CA, RA, or their combination is at the transcriptional or translational level. The authors were requested to perform qPCR analysis of IL1b mRNA in the presence and absence of CA, RA, or its combination in both OSCC cell lines.

The authors are requested to edit all typos and errors in the manuscript.

Reviewer 2 Report

The present work “Therapeutic Effects of Perilla Phenols in Oral Squamous Cell Carcinoma” by Lee et al. addresses a topic of importance for clinical and basic research – the use of Perilla phenols as a treatment strategy for oral squamous cell carcinoma.

Abstract is well structured and informative. Introduction is concise and correct. It summarizes the recent knowledge for the antitumoral efficacy of Perilla extract. The figures and the table are well presented and informative.  Their presentation is detailed and logically developed. Details are correct. Data shown are adequate to the aim and highly informative.

The only two remarks are listed below:

1. Lines 40-41. “Hong demonstrated that PLE is capable of inhibiting anti-SARS-CoV-2 replication by inactivating the virion. [9].” The sentence needs rewriting.

2.  Even that the “Materials and Methods” section is very well described, the information concerning Caki-1 cells in “Cell Culture” section is missing. 

Minor editing of English language required.

Round 2

Reviewer 1 Report

The authors have carefully addressed all the comments of the reviewers.

Minor English editing is required